# REPRESENTATION DISENTANGLEMENT VIA REGULARIZATION BY CAUSAL IDENTIFICATION

## ABSTRACT

In this work, we argue modern deep representation learning models for disentanglement are ill-posed with collider bias behaviour; a source of bias producing dependencies between the underlying generating variables. Under the rubric of causal inference, we show this issue can be explained and reconciled under the condition of causal identification; attainable from data and a combination of constraints, aimed at controlling the dependencies characteristic of the causal graphical *collider* model encoding the data generation process assumptions. For this, we propose regularization by identification (ReI), a modular regularization engine designed to align the behavior of large scale models with the disentanglement constraints imposed by causal identification. Empirical evidence on standard benchmarks demonstrates the superiority of ReI in learning disentangled representations in a variational framework. In a real-world dataset we additionally show that our framework results in interpretable representations robust to out-of-distribution examples and that align with the true expected effect from domain knowledge.

## 1 INTRODUCTION

One of the principal objectives of learning representations has been that of detecting measurement features that represent the qualitative and quantitative characteristics of the underlying physical processes being sensed. Most of the times sensing as dictated for example by the Nyquist rate Shannon (1948) acquires sufficient information for detection but leaves potentially unnecessary and redundant information on its measurements. Ideas to reduce such redundancies by representing information as concepts, patterns or features to achieve an economy of information have been the focus of study since its early days Pearson (1901). Standard variational formulations for representation learning such as the variational autoencoder (VAE) Kingma & Welling (2013) and denoising diffusion probabilistic models (DDPM's) Ho et al. (2020) have been among the most popular methods, very recently. Problem with these group of methods is their focus on learning approximations of the true marginal data distributions without any guarantees on the imposed representation priors to model the true underlying generative mechanisms Khemakhem et al. (2020). This not only disconnects the learned latent representations from real-meaning, obfuscating explainability of the generative process Lake et al. (2017) and counterfactual reasoning Pearl (2019), but also invokes problems of fairness and robustness to out-of-distribution (OOD) examples D'Amour et al. (2020).

Recent trends Bengio et al. (2013); Higgins et al. (2018); Locatello et al. (2019); Van Steenkiste et al. (2019); Khemakhem et al. (2020) are in consensus that disentanglement of the generating factors leads to increased robustness, explainability and fairness. The most widely used definition of disentangled representations assumes the set of underlying generative factors that explains the data has a one-to-one correspondence between each factor and a single (or a subset of) dimension(s) of the learned latent representations Bengio et al. (2013); Higgins et al. (2016); Chen et al. (2018); Eastwood & Williams (2018); Träuble et al. (2021); Roth et al. (2022). Recent efforts along this line of work, include unsupervised methods that rely on encoder heuristics to control the information bottleneck properties for disentanglement. Among the most popular methods includes $\beta$-VAE Higgins et al. (2016), Annealed VAE Burgess et al. (2018), Factor VAE Kim & Mnih (2018), DIP-VAE Kumar et al. (2018) all imposing specific structure in the latent prior through modifications of the Kullback-Leibler divergence (KL) term. The work of Yang et al. (2023) instead minimizes the mutual information between latent representations of an autoencoder and uses it in a guided denoising diffusion framework. These unsuperved methods, rely in general on a careful tunning of

the encoder hyper-parameters to preserve the desirable features in the data while destroying features of no particular interest (i.e., those from nuisance factors). Weakly-supervised methods on the other hand, have been shown to facilitate some form of disentanglement. Mitrovic et al. (2020) proposes a method that consists in learning representations explaining the causal data generation mechanisms by promoting invariance to augmented data transformations, this under the principle of independent causal mechanism Peters et al. (2016; 2017). The goal of invariant risk minimization (IRM) of Arjovsky et al. (2019); Rojas-Carulla et al. (2018), on the other hand, is to find representations that produce predictions invariant to environment contexts. Bouchacourt et al. (2018) proposes learning representations by grouped observations (i.e., a factor of variation shared between observations within a group) and uses a multi-level VAE for learning group representations as a generalization to i.i.d. assumptions. Locatello et al. (2020) demonstrates that disentangled representations can be obtained under weak-supervision when pairs of measurements share a factor of variation. Their approach modifies the $\beta$-VAE objective by enforcing similarities between the shared generative factors of variation and a decoupling of those uncommon. Worth noting is Träuble et al. (2021); Roth et al. (2022), whose findings extend disentanglement to cases with correlated factors; a problem that affects robustness to OOD examples D'Amour et al. (2020).

In light of these works, the breadth of efforts in this research focus on learning representations that disentangle the underlying generative factors of variation. Here, we follow the line of work of Suter et al. (2019); Locatello et al. (2020) where the framework for disentanglement is connected to the underlying causal mechanisms explaining the data generation process. Our main contributions are:

- Provide a connection between the definition of disentanglement and causal identification constraints which are informed by graphical causal models that encode the underlying data generation process.
- We argue modern approaches for learning disentangled representations exhibit collider-bias behavior. This is a type of bias, characterized under graphical causal models by a collider structure, that explains the appearance of conditional associations between the generating factors, even when these are in reality unrelated, thus producing entanglement.
- "Regularization by Identification" (ReI) a modular regularization engine designed to align the behavior of large scale DL models with causal identification constraints. This, enforces disentanglement by controlling dependencies between the underlying generating factors.
- A variational inference reformulation of the VAE representation learning problem (i.e., the ELBO) to achieve disentanglement by imposing ReI under the collider graphical model.
- Provide empirical evidence from both disentanglement benchmarks and real-world datasets showing the potential of ReI to produce representations that: (1) disentangle the effects of the generating factors with results well aligned with true expected behavior from domain knowledge that support interpretation and understanding and (2) are robust in the presence of out-of-distribution examples.

## 2 LEARNING DISENTANGLED REPRESENTATIONS

*Generative Representation Learning:* Consider observations $\mathbf{x} \in \mathcal{X} \subseteq \mathbb{R}^M$ drawn from distribution $\sim p^*(\mathbf{x})$. The goal of generative models for representation learning is to find encoders $q_{\boldsymbol{\theta}} : \mathcal{X} \to \mathcal{Z}$ that produce latent representations $\mathbf{z} \in \mathcal{Z} \subseteq \mathbb{R}^d$ distributed according to some prior distribution $p(\mathbf{z})$, that along with a generator $p_{\phi} : \mathcal{Z} \to \mathcal{X}$ marginally approximates the input data distribution.

### 2.1 CAUSAL INFERENCE BACKGROUND

**Definition 1.** *Directed Acyclic Graphs:* In the context of causal inference, the data generation process is represented by a directed acyclic graph (DAG) Pearl (1995). A DAG is a graphical model with domain variables represented as nodes, directed edges (i.e., arrows) expressing directional dependency relationships between variables. DAG's operate under the Markov compatibility property which states that the joint distribution $p$ is compatible with a DAG $G$ or that $G$ represents $p$ if it admits the decomposition

$$p(x_1, ..., x_n) = \prod_i p(x_i | pa_i). \tag{1}$$

Variables $pa_i$ are the Markovian parents of node $x_i$ that belong to the minimal set of predecessors that renders $x_i$ independent of all its other predecessors; in other words that, $p(x_i | pa_i) =$

$p(x_i|x_1, ..., x_{i-1})$ Pearl (2010). Parents $pa$ and predecessors are defined along the arrows in the graph. For example, in $X \to Z \to Y$, $X$ is the only parent of $Z$, $Z$ is the parent of $Y$ and $\{X, Z\}$ is the list of predecessors of $Y$. We note that this Markov compatibility assumes a first order Markov process as defined in Eq.(1). Causal paths between input $X$ and outcome $Y$ consist of a sequence of arrows following the causal direction and represents causal dependencies. Non-causal paths, on the other hand, consist of a sequence of connections between variables that lack a direct cause-and-effect relationship and represent dependencies that may arise from non-causal influences. If such dependencies remain untreated they can produce biased estimates. Examples include confounding from shared causes or collider-bias arising by conditioning on a common effect.

Causal inference analysis provides the tools to establish causality by predicting the effects of interventions $p(y|do(x))$ from the assumed DAG $G$ and ordinary distributions over observations. The way by which this is accomplished is by removing the effects of any non-causal dependencies between the input and output. The DAG provides the mathematical language for expressing domain knowledge through transparent and testable assumptions about the underlying relationships between domain variables. Transparency enables analysts to discern whether the stated assumptions encoded qualitatively in the DAG $G$ are plausible Pearl (2019) (on scientific grounds). Testability provides graphical criteria to test for causal/non-causal dependencies between variables. If the non-causal influences can be removed, then it provides the rules to do so. At the core of these causal tools, is the $d$-separation criteria Geiger et al. (1990). It is graphical-based, in the sense that the structure of the DAG $G$ (i.e., edge connections and paths) encodes qualitatively the patterns of dependencies we should expect to find in the data. In addition, it provides the means to control for any dependencies through conditioning by an appropriate set of covariates $\mathbf{Z}$.

**Definition 2.** Variable sets $X$ and $Y$ are $d$-Separated (or blocked) by a set $\mathbf{Z}$ denoted as $(X \perp\!\!\!\perp Y | \mathbf{Z})_G$, if and only if, $\mathbf{Z}$ blocks all paths from nodes in $X$ to nodes in $Y$ Geiger et al. (1990); Pearl (1995). The two general graphical conditions for blocking dependencies are:

- In the paths $X \to m \to Y$ or $X \leftarrow m \to Y$ the node $m$ is in $\mathbf{Z}$, or
- there is a collider $X \to m \leftarrow Y$ where neither node $m$ nor its descendant is in $\mathbf{Z}$.

When no feasible set $\mathbf{Z}$ exists then we say that $X$ and $Y$ are not $d$-separated. When it does then we can control for dependencies by conditioning on $\mathbf{Z}$. Definition 2 explicates that the $d$-separation criteria provides the graphical test to determine the dependencies that exist in the system of variables outside of input $X$ and outcome $Y$, and the mechanisms to control for these dependencies.

**Theorem 1.** *Verma & Pearl (1990) Probabilistic implications of $d$-Separation. $(X \perp\!\!\!\perp Y | \mathbf{Z})_G$ implies conditional independence of $X$ and $Y$ given a set of variables $\mathbf{Z}$ (including $\mathbf{Z} = \{\emptyset\}$) in every distribution compatible with the encoded assumptions in DAG $G$, while absence of $d$-Separation implies the converse; a dependence in almost all distributions compatible with the DAG.*

The implications of Theorem 1 allow us to control the dependencies that exist between two variables if they can be $d$-separated. Likewise, these dependencies are guaranteed to remain controlled under all distributions compatible with the specified DAG. In Pearl Pearl (1995) it is established that this, allows us to identify the causal effects between two variables under a specified DAG $G$ and data.

**Definition 3.** *Causal effect Identification Pearl (1995).* The causal effect of $x$ on $y$ denoted as $p(y|do(x))$ is identifiable from DAG $G$ and data, if a set of variables $\mathbf{Z}$ that $d$-separates them exists.

The implications of identification state that dependencies found in the DAG can be controlled through a combination of $d$-separation constraints. If the causal effect is identifiable then this control is guaranteed to hold in every distribution compatible with the DAG, while also being computable from the joint distribution $p(x, y, \mathbf{Z})$ over the observables. In other words, the causal effect in the l.h.s. of $p(y|do(X = x)) = \sum_{\mathbf{Z}} p(y|x, \mathbf{Z})p(\mathbf{Z}))$ can be computed through the r.h.s. involving only standard distributions over the observations. Note the excellent overview of methods for causal identification in Ayem et al.. In the following, we make a connection between the use of causal identification and the possibility and means to disentangle the underlying data generative factors.

## 2.2 CONNECTION BETWEEN DISENTANGLEMENT AND CAUSAL IDENTIFICATION

*Learning Disentangled Representations by Causal Identification.* Given a DAG $G$ encoding our assumptions about the data generative process, the possibility to learn disentangled representations

from data, we propose, can be tested through $d$-separation of the underlying generating factors. Moreover, identification of the causal effect $p(\mathbf{x}|do(\mathbf{y}_c))$ for all $c$ provides the means to control for dependencies between the generating factors producing entanglements, through a combination of $d$-separation constraints and data drawn from the joint distribution over the observables.

Given a dataset whose generative process is transparently encoded as a DAG with nodes representing the generative variables and edges the assumed relationships between them, the proposition involves to first qualitatively *test* whether the generative variables can be $d$-separated relative to each other. From Theorem 1, the possibility to identify the causal effect of each generative variable through $d$-separation implies that the dependencies between the generative variables can be controlled in every distribution compatible with the DAG $G$. According to our proposition, this allows, the possibility to disentangle the generative factors from a DAG $G$ and data with guarantees that hold in every distribution that satisfies the DAG compatibility. In addition, from Theorem 1, in the absence of $d$-separation there will be a dependence between the generating factors in almost all distributions compatible with the DAG. Without control of these dependencies, and given that DL models are good at exploiting shortcuts Beery et al. (2018); Geirhos et al. (2020); Pezeshki et al. (2021) provides no guarantees that DL models will learn a representation with disentanglement characteristics. Our proposition here instead, provides first a graphical qualitative test to describe the dependencies between variables expected in the data generating process and second, the derivations required to control dependencies of the generating factors which are guaranteed to hold under all distributions compatible with the DAG. This control of dependencies under all distributions, will ultimately lead to imposing control over the shortcuts explored and exploited by the representation learning model and constrain entanglement between the generating variables.

## 2.3 Directed Acyclic Graph: Collider-based Data Generation Model

At the core of causal identification, is the reliance on a graphical DAG model encoding the data generation process to identify (if possible) causal effects, and consequently remove dependencies between the generating factors producing entanglement. Here, we propose a simple DAG model that yet, explains and reconciles the entanglement between the generative factor effects on data.

*Collider Data Generative Model.* We argue that the underlying data generative model for disentanglement has collider structure. Generating variables (i.e., causes) $\mathbf{y}_c, \forall c \in \{1, ..., n\}$ have directed arrows colliding at node $\mathbf{x}$ (i.e., common effect). The DAG representing this simple, yet generic structural model is illustrated in Fig.1

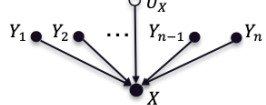

Figure 1: DAG $G$ encoding data generative process.

Variable realization $\mathbf{x} \in \mathbb{R}^M$ represents the sensor measurements, $\mathbf{y} \in \mathbb{R}^n$ with elements $\mathbf{y}_c, c \in \{1, ..., n\}$ are the underlying generating factors (permutations are possible as structure relationships are preserved). Nuisance factors (e.g., sensor noise, style), which are not relevant to the tasks we care about, are denoted by the unmeasured $\mathbf{u}_x \in \mathbb{R}^M$. Arrows emanating from $\mathbf{y}_c, \forall c$ to $\mathbf{x}$ (i.e., colliding at $\mathbf{x}$) are aligned with the causality of the generation mechanisms. In other words, there is causal precedence of the $\mathbf{y}_c$'s and they are a direct cause (i.e., implied by the arrow direction) of the common effect $\mathbf{x}$.

Inspection of Fig. 1 reveals potential dependencies arising from the structural connections $\mathbf{y}_c \to \mathbf{x} \leftarrow \mathbf{y}_j$ with $j \neq c$, indicative of a collider; a source of potential bias Berkson (1946); Kim & Pearl (1983); Pearl (2009). This bias occurs when two or more independent variables have a direct causal influence on a variable as they will become associated when conditioning on (e.g., observing) the common effect $\mathbf{x}$. This spurious conditional association between the generating factors is a source of entanglement if it remains without control. For example, consider the case of $C = A + B$ with $A$ and $B$ independent. When variable $C$ is observed (e.g., $C = 4$) variables $A$ and $B$ become conditionally correlated as knowledge of one informs the value of the second. Intuitively, this phenomenon occurs as information on one of the causes makes the other causes involved more or less likely given that the consequence has occurred (i.e., the explaining away effect Pearl (1988)); even when the causes are independent Pearl (1995). Such behavior has been observed in the context of data-based models also, but has been rather explained as a susceptibility to exploit shortcuts Beery et al. (2018); Geirhos et al. (2020); Pezeshki et al. (2021). Here, we argue such behavior can be explained as collider-bias from a collider structured data generative process.

Controlling for conditional dependencies between the generating factors and a single $\mathbf{y}_c$ in the collider model, requires finding the set that $d$-separates them. Inspection of Fig.1 reveals that $(\mathbf{y}_c \perp\!\!\!\perp \mathbf{x}|\mathbf{Z})_G$ with $\mathbf{Z} = \{\mathbf{y}_j : j \neq c\} \cup \{\mathbf{u_x}\}$. Causal identification $p(\mathbf{x}|do(\mathbf{y}_c))$, for each $c$, is thus given as:

$$p(\mathbf{x}|do(\mathbf{y}_c)) = \sum_{\mathbf{y}_{-c},\mathbf{u}_x} p(\mathbf{x}|\mathbf{y},\mathbf{u}_x)p(\mathbf{y}_{-c},\mathbf{u}_x) = \mathop{\mathbb{E}}_{p(\mathbf{y}_{-c},\mathbf{u}_x)}[p(\mathbf{x}|\mathbf{y},\mathbf{u}_x)] \qquad (2)$$

i.e., conditioning over $\mathbf{y}_{-c} = \{\mathbf{y}_j : j \neq c\}$ which denotes the generative factors in the system except $\mathbf{y}_c$. Full derivation of the expression in Eq.(2) can be found in Appendix B. Note that controlling for dependencies between the generating variables requires conditioning, which in turn requires some form of measurements of both $\mathbf{y}$ and $\mathbf{u_x}$. Identification is possible thus, given supervisory signals, or a weak-form of it Shu et al. (2019) for both $\mathbf{y}$ and $\mathbf{u_x}$. In cases when no measurements or proxies are available, dependencies between the involved variables that are compatible with the specified DAG will be present and representations will in turn present entanglements between the effects of such dependencies. For example, if variable $\mathbf{u_x}$ is unmeasured, opens any of the paths $\mathbf{y}_c \rightarrow \mathbf{x} \leftarrow \mathbf{u_x}$, $\forall c \in \{1,...,n\}$ leaving any plausible association between $\mathbf{y}_c$ and $\mathbf{u_x}$ without control. Similarly, the paths $\mathbf{y}_c \rightarrow \mathbf{x} \leftarrow \mathbf{u_x} \rightarrow \mathbf{x} \leftarrow \mathbf{y}_{-c}$ that remain without control, introduce entanglements between $\mathbf{y}_c, \mathbf{u_x}$ and $\mathbf{y}_{-c}$. Such collider model thus explains that DL models are free to exploit any of the unbounded number of models compatible with the specified DAG Jaber et al. (2018) unless some form of control through supervision or a weak form of it is available.

In addition to explaining entanglement effects and the means to control them, the collider DAG model is compatible with the independent factors model whose joint distribution admits factorizations of the form $p(\mathbf{y}) = \prod_c p(\mathbf{y}_c)$ as in Bouchacourt et al. (2018); Locatello et al. (2020); Khemakhem et al. (2020) and also of its correlated factors relaxation, as in Träuble et al. (2021). In causal-based disentanglement approaches, graphical DAG models have mostly focused on confounding structures characterized by pairs of generating variables that have a common cause Suter et al. (2019); Lu et al. (2021). The behaviour of such structures is fundamentally different from those of the collider structure. The former, does not deal with dependencies between the underlying generating factors from conditioning on the common effect. Thus, they are susceptible to this problem, notably in cases when datasets are not generated by i.i.d. factors, as in real-world scenarios. As a remark, Zhang et al. (2021) has explored this collider problem in the context of adversarial robustness.

## 2.4  REGULARIZATION BY CAUSAL IDENTIFICATION (ReI)

Representation learning models operate with the objective of fitting the joint distribution over the observations while simultaneously imposing a prior with desired structural characteristics. The Markov property of DAG's further constrains the unbounded number of plausible models that can fit the joint distribution to only those that are compatible with the specified DAG. ReI further restricts such models to those that control for entanglement through a reformulation of the learning problem. Such reformulation, imposes causal effect identification constraints of the generating factors to control for dependencies graphically encoded by a DAG $G$ as in Eq.(2).

*Regularization by Identification (ReI)*: a modular regularization engine designed to align the behavior of DL models by imposing generative factor disentanglement constraints through causal identification.

ReI aligns approximations that fit data distributions with disentanglement constraints by causal identification, which reformulates the learning problem as defined by:

$$\mathcal{L}(\boldsymbol{\theta};\mathbf{x}^{(i)},\mathbf{y}^{(i)}) = \underbrace{\mathcal{L}_\ell(\boldsymbol{\theta};\mathbf{x}^{(i)},\mathbf{y}^{(i)})}_{\text{Data}} + \lambda \underbrace{\mathcal{L}_\rho(\boldsymbol{\theta};\mathbf{x}^{(i)},\mathbf{y}^{(i)})}_{\text{ReI}} \qquad (3)$$

with $\lambda > 0$ being the regularization strength, $\boldsymbol{\theta} \in \boldsymbol{\Theta}$ a point of the space of DL models and, $\{\mathbf{x}^{(i)} \in \mathbb{R}^M, \mathbf{y}^{(i)} \in \mathbb{R}^n\}_{i=1}^N$ are the observations. The first term $\mathcal{L}_\ell$ is the likelihood function while the second $\mathcal{L}_\rho$ corresponding to ReI aligns the latent representation with disentanglement constraints imposed by causal identification (i.e., data + ReI) of $p(\mathbf{x}|do(\mathbf{y}))$.

ReI is different from the weakly-supervised setting of Mitrovic et al. (2020); Peters et al. (2017); Arjovsky et al. (2019) which aim at finding the generating mechanisms by imposing some form of invariance to real or augmented data variability. Or to Bouchacourt et al. (2018); Locatello et al. (2020); Träuble et al. (2021) imposing invariance to shared generative factors between at least pairs of observations while keeping those detected as varying, free. One additional property of our work is

that the encoder is set to produce latent vectors the same size as the inputs $\mathbf{x} \in \mathbb{R}^M$, in similarity to Ho et al. (2020). This design choice is made to avoid dependence on the information bottleneck principle while aiming at yielding representations suitable for interpretation.

## 2.5 Reformulation of the VAE to impose ReI for Disentanglement

The variational inference learning problem of the VAE in Kingma & Welling (2013) optimizes the evidence lower bound (ELBO) to approximate the true posterior $p^*(\mathbf{z}|\mathbf{x})$ given measurements $\{\mathbf{x}^{(i)}\}_{i=1}^N$. The ELBO can be formulated by two terms: a likelihood term $\mathcal{L}_\ell(\boldsymbol{\theta}; \mathbf{x}^{(i)}) = \mathbb{E}_{q(\mathbf{z}|\mathbf{x}^{(i)})} \left[\log p(\mathbf{x}^{(i)}|\mathbf{z})\right]$ and a regularizer given by the KL divergence as $\mathcal{L}_\rho(\boldsymbol{\theta}; \mathbf{x}^{(i)}) = D_{KL}(q(\mathbf{z}|\mathbf{x}^{(i)})||p(\mathbf{z}'))$. In the standard VAE, the prior $p(\mathbf{z}')$ typically a standard Gaussian, is used on the approximate posterior. ReI can reformulate the ELBO in the VAE to impose disentanglement constraints through causal identification. The reformulated posterior with controlled dependencies between generating variables for disentanglement given the assumed collider DAG $G$ structure in Fig.1 is equivalent to:

$$p(\mathbf{z}|\mathbf{x}, do(\mathbf{y}_c)) = p(\mathbf{x}|\mathbf{z}) \mathop{\mathbb{E}}_{p(\mathbf{y}_{-c})} \left[p(\mathbf{z}|\mathbf{y})\right] / p(\mathbf{x}, \mathbf{y}_c) \tag{4}$$

involving the observables $\{\mathbf{x}^{(i)}, \mathbf{y}^{(i)}\}_{i=1}^N$. The adjustments in Eq.(4) blocks dependencies between variables $\mathbf{y}_c \to \mathbf{z} \leftarrow \mathbf{y}_{-c}$ when observing $\mathbf{x}$. The full identification derivation of Eq.(4) is included in the Appendix B. The corresponding ELBO with ReI regularization imposing disentanglement constraints by causal identification results in the reformulated regularizer given as:

$$\mathcal{L}_\rho(\boldsymbol{\theta}, \phi; \mathbf{x}^{(i)}, \mathbf{y}^{(i)}) = D_{KL}\left(q(\mathbf{z}|\mathbf{x}^{(i)}, \mathbf{y}_c^{(i)})|| \mathop{\mathbb{E}}_{p(\mathbf{y}_{-c})} \left[p(\mathbf{z}|\mathbf{y}^{(i)})\right]\right) \tag{5}$$

Full-derivation is included in Appendix C. Note that imposing the disentanglement constraints through causal identification affects only the ELBO regularizer. The likelihood function in learning problems remains, without modification in general. Given these characteristics, we term our method ReI; as the disentanglement constraints required for causal identification can be directly imposed as a regularizer.

## 2.6 Experiments on Benchmark Datasets

We include experiments that show and compare the performance of DL models with ReI against state of the art methods on the task of disentanglement. In this task, the compared methods of Kingma & Welling (2013); Higgins et al. (2016; 2018); Kim & Mnih (2018); Chen et al. (2018); Locatello et al. (2020) are evaluated and in addition we also include experiments in the non-idealized setting of correlated generating factors as in Träuble et al. (2021); Roth et al. (2022). We use the VAE+ReI described in Section 2.5 with causal identification derived from the collider DAG in Fig.1. The datasets used are the standard ML benchmarks used for learning disentangled representations: Shapes3D Kim & Mnih (2018), dSprites Higgins et al. (2016) and MPI3D Gondal et al. (2019).

### 2.6.1 Explicit Regularizations that Control Collider Behavior Through Causal Identification Do Better at Standard Disentangled Metrics

The metric of disentanglement performance used here is DCI (Disentanglement, Completeness, Informativeness) scores Eastwood & Williams (2018). DCI has been established as the most widely accepted metrics of disentanglement performance Locatello et al. (2019; 2020); Träuble et al. (2021); Roth et al. (2022). DCI evaluations are performed in synthetic datasets generated by either independent or by correlated factors. For the later, we use the extensions in Träuble et al. (2021); Roth et al. (2022) to correlate one, two and three generative factor pairs (where applicable) and one to all factors (1-to-all). We report the average metric and standard deviation (in square brackets) computed over 10 seeds and present the results in Tables 1, 2 and 3 for all three datasets.

Here, we see that the DCI performance degrades throughout all datasets as the number of correlated pairs increases for most of the methods compared. These do not seem to be well equipped to handle an increasing number of correlated generating factors. The observed degradation can be explained by the behavior of a collider where factors without control introduce dependencies between them producing entanglements. The severity of such dependencies depends on the number of factors that remain unadjusted for. The fact that the compared methods do not address this collider behavior explicitly, explains the lower performance as the number of correlated pairs increases. By explicitly addressing

Table 1: DCI-Disentanglement Performance Comparison on **dSprites** Higgins et al. (2016)

| Method | Uncorr. | Pairs: 1 | Pairs: 2 | 1-to-All |
|---|---|---|---|---|
| $\beta$-VAE Higgins et al. (2016) | 32.3 [8.7] | 9.4 [2.8] | 7.8 [2.5] | 11.3 [3.9] |
| Factor-VAE Kim & Mnih (2018) | 25.2 [7.9] | 13.1 [6.7] | 14.1 [4.2] | 14.4 [3.4] |
| $\beta$-TCVAE Chen et al. (2018) | 31.3 [5.8] | 23.9 [0.9] | 11.3 [5.2] | 20.3 [6.1] |
| Annealed-VAE Burgess et al. (2018) | 39.2 [3.1] | 14.8 [2.2] | 8.7 [2.4] | 14.2 [0.7] |
| $\beta$-VAE + HFS Roth et al. (2022) | 49.2 [15.1] | 19.2 [2.9] | 17.5 [12.3] | 15.9 [2.9] |
| $\beta$-TCVAE + HFS Roth et al. (2022) | 53.3 [9.2] | 26.2 [3.0] | 27.5 [10.9] | 24.5 [4.2] |
| VAE + ReI | **87.5** [7.1] | **88.2** [7.4] | **88.1** [7.2] | **89.4** [6.1] |

Table 2: DCI-Disentanglement Performance Comparison on **Shapes3D** Kim & Mnih (2018)

| Method | Uncorr. | Pairs: 1 | Pairs: 2 | Pairs: 3 | 1-to-All |
|---|---|---|---|---|---|
| $\beta$-VAE Higgins et al. (2016) | 70.3 [9.2] | 71.2 [8.9] | 51.6 [9.0] | 36.5 [4.9] | 36.3 [2.7] |
| Factor-VAE Kim & Mnih (2018) | 62.3 [13.6] | 70.8 [1.6] | 58.7 [5.5] | 46.1 [6.1] | 31.9 [6.2] |
| $\beta$-TCVAE Chen et al. (2018) | 77.4 [3.1] | 70.2 [5.6] | 63.4 [4.7] | 38.8 [11.4] | 51.9 [7.5] |
| Annealed-VAE Burgess et al. (2018) | 62.1 [2.6] | 55.7 [7.3] | 30.8 [6.1] | 36.2 [5.2] | 23.1 [4.3] |
| $\beta$-VAE + HFS Roth et al. (2022) | 91.8 [17.9] | 79.8 [3.7] | 67.3 [5.1] | 48.7 [5.0] | 63.4 [3.2] |
| $\beta$-TCVAE + HFS Roth et al. (2022) | 86.3 [3.6] | 75.6 [2.6] | 66.3 [7.7] | 51.7 [3.8] | 61.4 [7.9] |
| VAE + ReI | **95.9** [5.4] | **96.6** [3.4] | **96.3** [1.9] | **96.1** [2.8] | **95.8** [6.3] |

this type of bias by leveraging the power of causal models, the performance of ReI remains more or less invariant to the number of correlation pairs and their strength, as long as causal identification is possible. This is one of the main benefits of ReI, which of course comes at the cost of requiring a supervisory signal (labels in these cases) to identify the effects of the generating factors.

In contrast, the MPI3D dataset comes from real images captured from a moving robotic arm. The relatively lower performance of all methods on this dataset can be attributed to the fact that it comes from a real-world scenario with several unmeasured factors of variation. The images in MPI3D are obtained from three different cameras each affected by sensor noise, blur, illumination changes from view. The unconstrained setting, results in representations entangled from conditional dependencies with such variables $\mathbf{u_x}$ in Fig.1. In this sense, the collider DAG structure offers an explanation and the conditions for control. Control can be performed through additional experiments, gathering additional observations about $\mathbf{u_x}$, or assuming a parametric form to provide supervision. This, is an advantage over the methods compared, where explanations and ways to remediate, remain a black-box.

## 3 EXPERIMENTS ON REAL-WORLD DATASET

Experiments were conducted in spectroscopic applications, specifically using data from a laser induced breakdown spectroscopy (LIBS) instrument. LIBS is a remote sensing technology used to predict the chemical composition of geomaterials (e.g., rocks, soil) based on its signatures. On Mars, the ChemCam LIBS based instrument is equipped with a 1064nm laser and ultraviolet, visible and near infrared band spectrometers; which altogether is capable of collecting the sample's spectral signatures between 240-905nm. Focus here, is applying the ReI framework directed towards: (1) representation disentanglement, (2) prediction and (3) transfer. (1) learns representations characteristic of specific chemical elements. (2) uses the learned representations to predict chemical content, while (3), tests for robustness to dataset shifts by training data from Earth in a controlled setting while deployment is in the wild on Mars. Additional details are included in Appendix E.1

### 3.1 REPRESENTATION DISENTANGLEMENT

The abilities of ReI for learning disentangled representations were evaluated here. Training utilizes example pairs $\{\mathbf{x}^{(i)}, \mathbf{y}^{(i)}\}_{i=i}^{N}$ of LIBS signal $\mathbf{x} \in \mathbb{R}^{5485}$ measurements and corresponding true chemical composition $\mathbf{y}$. Percentages $\mathbf{y}_c$ represent % oxide composition for $c \in \{1, ..., 11\}$ indexing $\{SiO_2, TiO, Al_2O_3, FeO_T, MgO, MnO, CaO, Na_2O, K_2O, CO_2, H_2O\}$ and sensor noise is

Table 3: DCI-Disentanglement Performance Comparison on **MPI3D** Gondal et al. (2019)

| Method | Uncorr. | Pairs: 1 | Pairs: 2 | Pairs: 3 | 1-to-All |
|---|---|---|---|---|---|
| $\beta$-VAE Higgins et al. (2016) | 25.9 [7.9] | 18.3 [2.4] | 23.7 [1.3] | 11.3 [0.5] | 11.2 [2.0] |
| Factor-VAE Kim & Mnih (2018) | 26.6 [2.0] | 22.8 [2.8] | 28.2 [1.5] | 11.0 [0.9] | 13.8 [0.8] |
| $\beta$-TCVAE Chen et al. (2018) | 27.3 [1.0] | 20.9 [0.7] | 22.8 [1.4] | 11.1 [1.7] | 14.5 [1.5] |
| Annealed-VAE Burgess et al. (2018) | 11.4 [1.3] | 12.3 [1.9] | 11.9 [0.4] | 10.7 [1.2] | 13.1 [0.8] |
| $\beta$-VAE + HFS Roth et al. (2022) | 32.9 [3.2] | 29.2 [2.2] | 27.3 [0.6] | 13.8 [1.3] | 15.7 [1.2] |
| $\beta$-TCVAE + HFS Roth et al. (2022) | 32.6 [3.4] | 28.6 [4.1] | 29.1 [0.7] | 11.4 [ 3.9] | 15.2 [1.3] |
| VAE + ReI | **73.5** [5.5] | **72.6** [7.2] | **74.3** [6.3] | **71.9** [3.2] | **73.5** [4.5] |

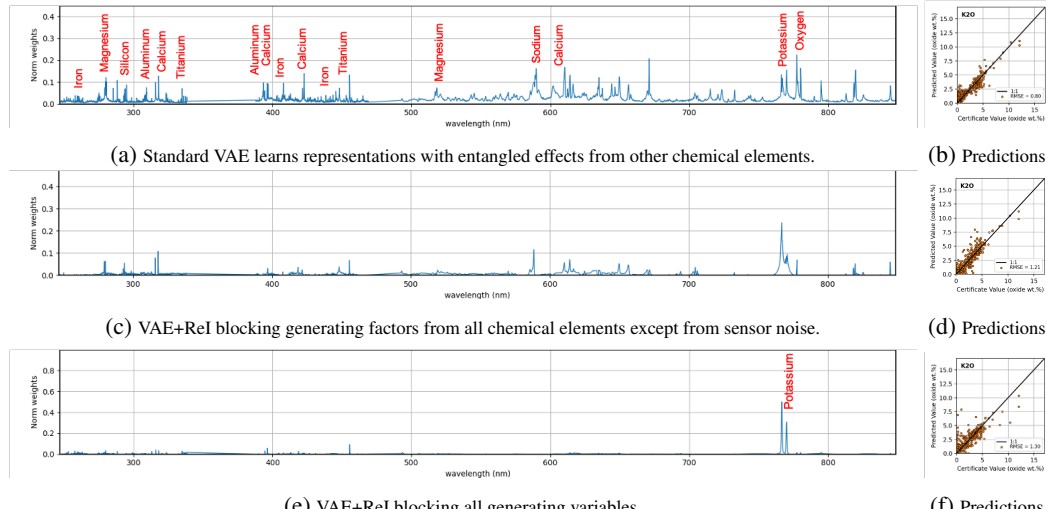

(a) Standard VAE learns representations with entangled effects from other chemical elements.

(b) Predictions

(c) VAE+ReI blocking generating factors from all chemical elements except from sensor noise.

(d) Predictions

(e) VAE+ReI blocking all generating variables.

(f) Predictions

Figure 2: Comparison of the learned representations for chemical oxide $K_2O$.

$\mathbf{u_x} \in \mathbb{R}^{5485}$. Qualitative evaluations of the representations from ReI derived from the collider DAG in Fig.1 were performed in light of the known characteristic spectral response of each chemical oxide. We used an MLP architecture and compared the representations in three cases: (1) the standard VAE, (2) VAE+ReI with all factors identified except for sensor noise and (3) ReI with all generating factors identified. Training used 585 reference targets under leave one out while testing was done on the target left out until all are covered. Additional implementation details are included in Appendix E.2.

A representative example on the learned representations corresponding to chemical oxide $K_2O$ is shown in Fig.2. These were generated by sampling from $q(\mathbf{z}|\mathbf{x}^{(i)}, \mathbf{y}_c^{(i)})$ with $\mathbf{y}_c$ as composition of $K_2O$ and averaging over $L = 100$ samples. Figs.2a, 2c and 2e shows the learned representations for $K_2O$ in: (1) VAE, (2) VAE+ReI with sensor noise $\mathbf{u_x}$ uncontrolled and (3) VAE+ReI with control for all generating factors. The vertical axis of each plot shows the normalized magnitude and the horizontal axis represents spectral wavelength importance. Figs.2b, 2d and 2f illustrate the corresponding prediction performance $\tilde{\mathbf{y}}_c$ of the three cases using the representations $\mathbf{z}$ along with a trained linear prediction head. Prediction performance by looking into point distribution along the 1:1 line and as measured by the root mean squared error (RMSE) shows similar performances in all three cases; with a marginal advantage of the standard VAE (i.e., (1) 0.8, (2) 1.21, (3) 1.30). However, these all come from distinct learned representations with key observations supporting evidence of collider behavior. First, note that $K_2O$ (Potassium oxide) is known and expected to respond to wavelengths around $\sim 770$nm as labeled in Fig.2a, illustrating the ground truth expected spectral responses of a variety of chemical elements. The standard VAE in Fig.2a resulted in a representation with spectral peaks deemed important spread throughout the entire spectrum. This is indicative of conditional dependencies between the generative factors $\mathbf{y}_{-c} = \{\mathbf{y}_j : j \neq c\}$ and $K_2O$ through the path $\mathbf{y}_{-c} \rightarrow \mathbf{x} \leftarrow \mathbf{y}_c$. Fig.2c in contrast shows the resulting representation obtained by VAE+ReI with control for dependencies between the generative factors except for those from sensor noise $\mathbf{u_x}$. Although most of the wavelengths previously deemed important were flattened, some small

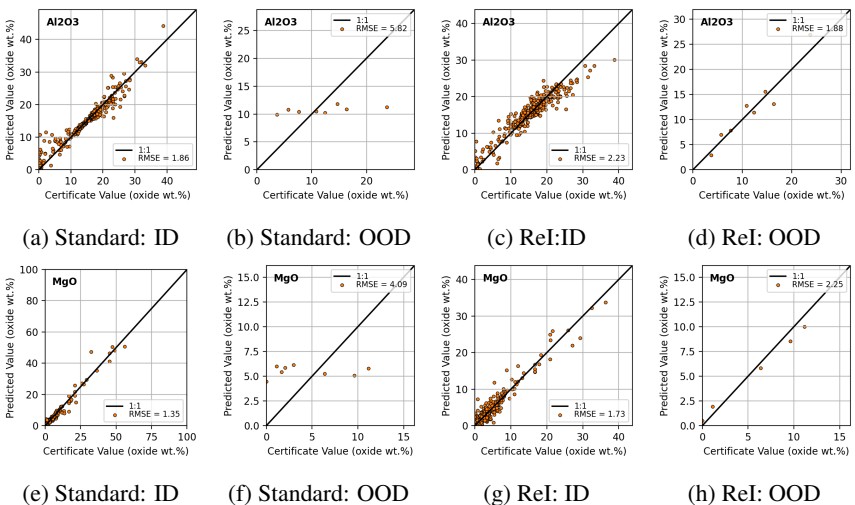

Figure 3: Performance comparison of the learned representations in out-of-distribution transfer.

spectral peak patterns from Fig.2a persisted. This, due to conditional associations between the paths $\mathbf{y}_{-c} \to \mathbf{x} \leftarrow \mathbf{u_x}$ and $\mathbf{u_x} \to \mathbf{x} \leftarrow \mathbf{y}_c$. Finally, Fig.2e illustrates the representation by VAE+ReI with control for all generative factors. Most wavelengths were brought down to zero except for the two strong peaks at $\sim 770$nm in alignment with the expected spectral response for $K_2O$. Identification thus produced representations well aligned with the expected effects of the generating factors.

The empirical evidence provided supports our claim that standard generative representation learning models ill-suffer from collider bias. Note that downstream tasks, such as the prediction of in-distribution examples and without visualizations of the learned representations as exemplified by Figs.2b, 2d and 2f can obscure the aforementioned illness. However, illustrations of the representations of the effects of generating factors clearly shows evidence of this problem, with effects supporting collider behavior. These findings, thus provide a plausible alternative explanation to the spurious association problems between factors found in Geirhos et al. (2020); Pezeshki et al. (2021), to fairness Zhao et al. (2017), and provide a venue for analysis and remediation through causality as viewed by Pearl (2010) and tackled here by ReI. We would like to note also that the learned representations from the VAE+ReI are amenable for interpretation while also explain the effects of generating factors as they relate to the effects of the measuring apparatus, supporting understanding.

### 3.2 PREDICTION AND TRANSFER

Quantitative comparisons on the robustness to dataset shifts is performed here. Dataset shifts originate here by training from data collections of LIBS from targets on Earth in a laboratory setting while deployment occurs in the wild on Mars. Clegg et al. (2017) found that the Martian environment has effects that shift the distribution of measurements relative to Earth. This task, then seeks to investigate the transferability of the learned representations in the presence of OOD shifts.

Example representative results are included in Fig.3 which shows true versus prediction plots for two element oxides $Al_2O_3$ and MgO. Four leftmost Figs.3a,3e,3b,3f corresponds to performance results from the standard VAE+FC linear head whereas the rightmost four Fig.3c,3g,3d,3h shows those from VAE+ReI+FC. Figs. 3a,3e,3c,3g show that both VAE and VAE+ReI present similar performance for in-distribution example testing (under leave one out), with the VAE being marginally better in terms of RMSE. In contrast, Figs.3b,3f,3c,3g shows significant differences in performance in the OOD cases. VAE+ReI presents better behaved performance and outperforms by larger margins compared to the VAE. Note that even though the VAE presents an advantage over VAE+ReI for in-distribution performance, that this is not the case for OOD examples. The disentanglement provided by VAE+ReI shows better robustness against OOD examples. This behavior, is consistent with findings by Tsipras et al. (2018), where highly predictive non-robust features in the data tend to reduce learner performance when presented with OOD examples.

## 4 CONCLUSIONS

In this work, we proposed ReI: a regularization method that aligns DL models to domain knowledge by leveraging the DAG. We argued that standard disentangled learning models are ill-biased by collider behaviour and showed supporting empirical evidence of this. In a variational framework, we showed how analysis of the DAG under the lens of causality can be used to control for collider bias via ReI in representation learning problems. Empirical evidence shows ReI is capable of learning the effects between the generating factors and the sensor, removing collider bias, producing representations in disentangled form, generalizable to OOD example cases and supporting interpretation of both factor effects and manipulations of these for sampling posterior generation.

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

## A  BACKGROUND

### A.1  THE RULES OF $do$-CALCULUS

The axioms branded under the $do$-calculus are presented here.

In terms of notation, in a DAG $G$, $G_{\overline{X}}$ and $G_{\underline{X}}$ denote, respectively, the graphs obtained by deleting the incoming and outgoing arrows at node $X$.

The rules of interventional $do$-calculus according to Pearl are given as Pearl (1995; 2010):

"**Rule 1** (Insertion/deletion of observations):

$$p(y|\hat{x}, z, w) = p(y|\hat{x}, w) \quad \text{if} \quad (Y \perp\!\!\!\perp Z)|X, W)_{G_{\overline{X}}}$$

**Rule 2** (Action/observation exchange):

$$p(y|\hat{x}, \hat{z}, w) = p(y|\hat{x}, z, w) \quad \text{if} \quad (Y \perp\!\!\!\perp Z)|X, W)_{G_{\overline{X}\underline{Z}}}$$

**Rule 3** (Insertion/deletion of actions):

$$p(y|\hat{x}, \hat{z}, w) = p(y|\hat{x}, w) \quad \text{if} \quad (Y \perp\!\!\!\perp Z)|X, W)_{G_{\overline{X}\overline{Z}}}"$$

These graphical rules encompass the foundational principles of the $do$-calculus Pearl (1995). By analyzing the DAG and employing these rules, it becomes possible to characterize the effects of interventions $do(x)$ in terms of ordinary probability distributions of observations. This process, known as identification in the context of causal inference, serves as the primary analytical tool for elevating relationships between variables from mere correlation to causation.

### A.2  A SIMPLE, BUT CLARIFYING EXAMPLE

As an example, consider the learning problem $p(y|x)$ (e.g., label prediction $y$ from observations $x$) where we observe three variables $x, y, z$ from a joint distribution $p(x, y, z)$. In the absence of a data generation process model, this problem can be expressed as $p(y|z) = \sum_z p(x, y, z)/p(x)$. A DL model for solving this problem would learn a function that fits the joint distribution over the observed variables. However, there are numerous models that can fit this distribution, such as $p(x, y, z) = p(y|z)p(z|x)p(x)$, $p(x|z)p(z|y)p(y)$, $p(y|x, z)p(x)p(z)$, and many more combinations of variables. The issue arises because each of these decompositions represents a specific generative model, and without any restrictions, the DL model is free to choose any of them. This can be problematic as there exists many models that do not align to domain knowledge and this can invoke a variety of additional problems. For example, DL models that lack interpretability or that fail at transfer in the presence of data distribution shifts. Imposing a data generative model through DAG's thus constrains the DL model to align its behavior with the specified DAG.

In the specific case example shown in Fig.4a, the DL model fitting the joint distribution over the observations would align with the generative model represented by the decomposition $p(x, y, z) = p(y|x, z)p(x|z)p(z)$. However, a potential issue arises as $z$ influences both the independent variable $x$ and the dependent variable $y$, leading to spurious correlations between variables and introducing

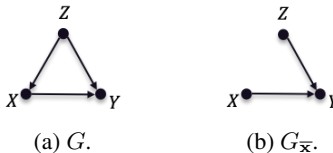

(a) $G$.    (b) $G_{\overline{\mathbf{x}}}$.

Figure 4: DAG example

a potential source of bias for the learning problem $p(y|x)$. Lifting this model from correlation to causation through the application of $do$-calculus rules, tackles this issue by analyzing the system in terms of interventions. Utilizing $do$-calculus, we can reformulate the learning problem as $p(y|\hat{x})$, and its identification can be expressed as:

$$p(y|\hat{x}) = \sum_z p(\hat{x}, y, z)/p(\hat{x}) \tag{6}$$

$$= \sum_z p(y|z, \hat{x})p(\hat{x}|z)p(z)/p(\hat{x}) \tag{7}$$

$$= \sum_z p(y|z, \hat{x})p(\hat{x})p(z)/p(\hat{x}) \tag{8}$$

$$= \sum_z p(y|z, x)p(z) = \mathop{\mathbb{E}}_{p(z)} p(y|z, x) \tag{9}$$

where Eq.(8) follows since conditioning on parent variables has no effect on interventions. Eq.(9) follows by noting that $x$ and $y$ are $d$-separated under Fig.4b. As such, we can exchange the action $p(y|\hat{x}, z)$ for the observation $p(y|x, z)$ as described by Rule 2 of the $do$-calculus. A DL model that learns by fitting $p(y|\hat{x})$ from Eq.(9) instead of $p(y|x)$ aligns with the causal data generation process encoded by the DAG. This alignment, helps alleviate potential sources of bias or dependencies arising from non-causal paths, that can be identified in the system through causal analysis of the DAG. In our proposed research, we are driven by the fundamental premise that causal models offer both superior disentanglement and representation of domain knowledge compared to mere correlations. To incorporate these causally derived constraints, we leverage derived distributions over interventions $p(y|\hat{x})$ and reformulate the learning problem accordingly.

## B   CAUSAL EFFECT IDENTIFICATION IN COLLIDER STRUCTURE

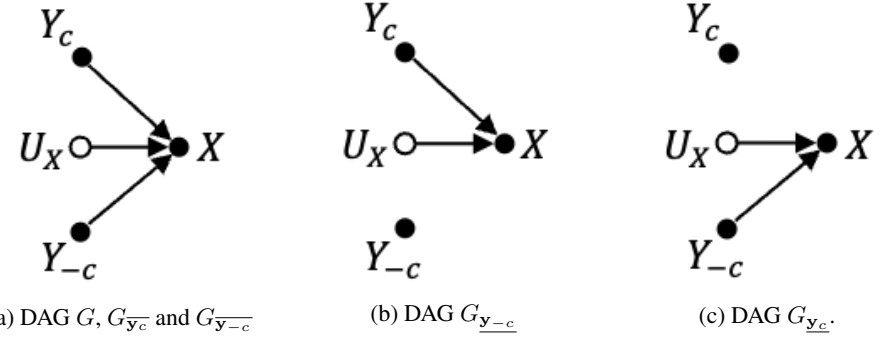

(a) DAG $G$, $G_{\overline{\mathbf{y}_c}}$ and $G_{\overline{\mathbf{y}_{-c}}}$    (b) DAG $G_{\underline{\mathbf{y}_{-c}}}$    (c) DAG $G_{\underline{\mathbf{y}_c}}$.

Figure 5: A DAG with colliders

Identification of the causal effects $p(\mathbf{x}|\hat{\mathbf{y}}_c)$ in Eq.(1) involves application of the rules of the $do$-calculus by leveraging the causal assumptions encoded in the DAG. This is used to convert probabilities of interventions to expressions involving only ordinary probabilities of observations. The DAG in Fig.5a is a representation of the data generative model of independent factors $p(\mathbf{y}) = \prod_{c=1}^{n} p(\mathbf{y}_c)$ assumed in Kingma & Welling (2013); Higgins et al. (2016; 2018); Kim & Mnih (2018) and a simplification of Fig.1. In this case however, with only three factor variables $\mathbf{y}_c, \mathbf{y}_{-c}, \mathbf{u}_x$. The DAG structure in

Fig.5a contains a collider at $\mathbf{x}$. A collider is represented in a DAG by a node where two or more arrows or paths converge. A collider, produces conditional associations between the generating factors $\mathbf{y}_c, \mathbf{y}_{-c}, \mathbf{u}_x$, even if they are not causally related. This phenomenon is known as collider bias or selection bias Berkson (1946); Kim & Pearl (1983); Pearl (2009). This needs to be accounted for, when training a DL model to learn from the joint distribution $p(\mathbf{x}, \mathbf{y}, \mathbf{u}_x)$ over the observations, otherwise it is prone to produce biased models. Again, we do this through derivations involving the effects of interventions $p(\mathbf{x}|\hat{\mathbf{y}}_c)$ in the system encoded by Fig.5a. Application of the law of total probability in Eq.(10) and the chain rule in Eq.(11) both valid under probabilities of interventions Pearl (2009) yields:

$$p(\mathbf{x}|\hat{\mathbf{y}}_c) = \sum_{\mathbf{y}_2, \mathbf{u}_x} p(\mathbf{x}, \hat{\mathbf{y}}_c, \hat{\mathbf{y}}_{-c}, \mathbf{u}_x)/p(\hat{\mathbf{y}}_c) \tag{10}$$

$$= \sum_{\mathbf{y}_{-c}, \mathbf{u}_x} p(\mathbf{x}|\hat{\mathbf{y}}_c, \mathbf{y}_{-c}, \mathbf{u}_x)p(\hat{\mathbf{y}}_c|\mathbf{y}_{-c}, \mathbf{u}_x)p(\mathbf{y}_{-c}, \mathbf{u}_x)/p(\hat{\mathbf{y}}_c) \tag{11}$$

$$= \sum_{\mathbf{y}_{-c}, \mathbf{u}_x} p(\mathbf{x}|\hat{\mathbf{y}}_c, \mathbf{y}_{-c}, \mathbf{u}_x)p(\mathbf{y}_{-c}, \mathbf{u}_x) \tag{12}$$

$$= \sum_{\mathbf{y}_{-c}, \mathbf{u}_x} p(\mathbf{x}|\mathbf{y}, \mathbf{u}_x)p(\mathbf{y}_{-c}, \mathbf{u}_x) = \mathbb{E}_{\mathbf{y}_{-c}, \mathbf{u}_x} \left[ p(\mathbf{x}|\mathbf{y}, \mathbf{u}_x) \right] \tag{13}$$

Note that we have used for easy of notation $\hat{\mathbf{y}}_c = do(\mathbf{y}_c)$. Eq.(12) follows by definition of the $do$ operator implying no effect on an intervention conditioned on any variables (i.e., $p(\hat{\mathbf{y}}_c|\mathbf{y}_{-c}) = p(\hat{\mathbf{y}}_c) = 1$). The later, follows from the definition of an intervention, where there is no uncertainty on the variable being intervened upon. Eq.(13) follows as Rule 2 for action/observation exchange is satisfied. In other words, given that $(\mathbf{X} \perp\!\!\!\perp \mathbf{Y}_c)|\mathbf{Y}_{-c}, \mathbf{U}_x)_{G_{\overline{\mathbf{y}_c}}}$ is $d$-separated in $G_{\underline{\mathbf{y}_c}}$ in Fig.5b allows the exchange from $p(\mathbf{x}|\hat{\mathbf{y}}_c, \mathbf{y}_{-c}, \mathbf{u}_x)$ to $p(\mathbf{x}|\mathbf{y}_c, \mathbf{y}_{-c}, \mathbf{u}_x)$ and since $\mathbf{y} = [\mathbf{y}_c, \mathbf{y}_{-c}]$ by our definition, completes the proof. Extensions to cases where $n > 2$ as in the generative model in Sec.2.1 follows trivially through the same derivation.

Abusing causal notation, we derive the identifiability conditions of the query $p(\mathbf{z}|\mathbf{x}, \hat{\mathbf{y}}_c)$ in Eq.(5) noting that all terms involved respect the causal direction.

$$p(\mathbf{z}|\mathbf{x}, \hat{\mathbf{y}}_c)$$
$$= p(\mathbf{x}|\mathbf{z}, \hat{\mathbf{y}}_c)p(\mathbf{z}|\hat{\mathbf{y}}_c)p(\hat{\mathbf{y}}_c)/p(\mathbf{x}, \hat{\mathbf{y}}_c) \tag{14}$$
$$= p(\mathbf{x}|\mathbf{z}) \mathbb{E}_{\mathbf{y}_{-c}, \mathbf{u}_x} \left[ p(\mathbf{z}|\mathbf{y}, \mathbf{u}_x) \right] /p(\mathbf{x}, \mathbf{y}_c) \tag{15}$$

Eq.(14) follows by application of the chain rule of probability. Deletion of actions from $p(\mathbf{x}|\mathbf{z}, \hat{\mathbf{y}}_c)$ follows by applying Rule 3, satisfied when $d$-separation $(\mathbf{X} \perp\!\!\!\perp \mathbf{Y}_c)|\mathbf{Z})_{G_{\overline{\mathbf{y}_c}}}$ is satisfied. By inspection of Fig.5c we see this is indeed the case. Also, the action $p(\hat{\mathbf{y}}_c)$ is by definition one and substituting the result of the conditional latent distribution $p(\mathbf{z}|\hat{\mathbf{y}}_c)$ in Eq.(13) completes writing an equivalent expression involving only ordinary probabilities of observations for the numerator. An expression for the denominator $p(\mathbf{x}, \hat{\mathbf{y}}_c)$ follows by adding $\mathbf{z}$ through the law of total probability and using the chain rule as $\sum_z p(\mathbf{x}, \hat{\mathbf{y}}_c|\mathbf{z})p(\mathbf{z})$. The action/observation exchange $\sum_z p(x, \mathbf{y}_c|\mathbf{z})p(\mathbf{z})$ then follows by checking if $(\mathbf{X} \perp\!\!\!\perp \mathbf{Y}_c)|\mathbf{Z})_{G_{\underline{\mathbf{y}_c}}}$ is satisfied; which is indeed the case by inspection of Fig.5b, completing the proof.

When at least one of the generating factors, such as $\mathbf{u}_x$ (e.g., sensor noise), remains unmeasured, it will leave several paths (e.g., $\mathbf{y}_c \leftrightarrow \mathbf{u}_x$, $\mathbf{y}_c \leftrightarrow \mathbf{u}_x \leftrightarrow \mathbf{y}_{-c}$) in the collider unblocked. This means, the causal effect of $\mathbf{y}_c$ on $\mathbf{z}$ cannot be identified uniquely, but only a relaxed relationship where $p(\mathbf{z}|\tilde{y}_c)$ may carry information correlations with both $\mathbf{u}_x$ and $\mathbf{y}_{-c}$. The strength of such correlations depends in this case on the energies of $\mathbf{u}_x$ relative to $\mathbf{z}$. But, overall the strength of these correlations has a direct impact on the severity of bias in DL models. This problem can be aggravated exponentially when the number of unmeasured generating factors increases. One of the main arguments in this research is that collider bias is prevalent in the majority of DL models designed to disentangle generative factors. These models often fall short in recognizing and effectively addressing this issue. We propose leveraging the power of causal models, specifically DAGs, to effectively incorporate a transparent and explicit model of the generative process. This integration aims to identify and mitigate the influence of colliders on disentanglement tasks. By leveraging DAGs, we can enhance the understanding and management of collider effects, improving the overall performance of disentanglement DL models.

## C VAE+ReI Reformulation: Alignment with Causal Collider Structure

Through ReI, we align the VAE framework, with the causal DAG of Fig.5a through a reformulation of the ELBO that accounts for the presence of a collider. The reformulation describes the learning problem in terms not of the ordinary posterior $q(\mathbf{z}|\mathbf{x}, \mathbf{y})$ but rather in terms of an interventional posterior $q(\mathbf{z}|\mathbf{x}, \hat{\mathbf{y}})$. Derivation of this reformulated ELBO in Eq.(6) follows the same steps as in Kingma & Welling (2013) starting with the Kullback-Leibler (KL) divergence.

In the case of the VAE, the likelihood term is given by Eq.(16) as:

$$\mathcal{L}_\ell(\boldsymbol{\theta}, \phi; \mathbf{x}^{(i)}) = \mathop{\mathbb{E}}_{q(\mathbf{z}|\mathbf{x}^{(i)})} \left[ \log p(\mathbf{x}^{(i)}|\mathbf{z}) \right] \tag{16}$$

and the regularizer $\mathcal{L}_\rho(\boldsymbol{\theta}, \phi; \mathbf{x}^{(i)})$ given in case of the standard VAE by the Kullback-Leibler (KL) divergence

$$\mathcal{L}_\rho(\boldsymbol{\theta}, \phi; \mathbf{x}^{(i)}) = D_{KL}(q(\mathbf{z}|\mathbf{x}^{(i)})||p(\mathbf{z})) \tag{17}$$

imposing a prior $p(\mathbf{z})$, typically a standard Gaussian, on the approximate posterior. The $\boldsymbol{\theta}, \phi$ are the parameters of the encoder and decoder models, respectively, and optimized over the training dataset. A scalar $\lambda$ is typically introduced as a multiplier in front of the r.h.s. of Eq.(17) as the regularizer strength balancing tradeoffs between the likelihood and priors, a parameter utilized by the $\beta$-VAE to promote the prior structure. Thes regularizer in Eq.(17) is reformulated by ReI to impose disentanglement constraints using the collider model structure shown in Fig.1. The steps of the full derivation are:

$$D_{KL}(q(\mathbf{z}|\mathbf{x}, \hat{\mathbf{y}}_c)||p(\mathbf{z}|\mathbf{x}, \hat{\mathbf{y}}_c)) = -\sum q(\mathbf{z}|\mathbf{x}, \mathbf{y}_c) \log p(\mathbf{z}|\mathbf{x}, \hat{\mathbf{y}}_c)/q(\mathbf{z}|\mathbf{x}, \mathbf{y}_c) \tag{18}$$

$$= -\mathop{\mathbb{E}}_{q(\mathbf{z}|\mathbf{x}, \mathbf{y}_c)} \{\log p(\mathbf{x}|\mathbf{z})\} - \sum q(\mathbf{z}|\mathbf{x}, \mathbf{y}_c) \log \mathop{\mathbb{E}}_{p(\mathbf{y}_{-c})} [p(\mathbf{z}|\mathbf{y})]/q(\mathbf{z}|\mathbf{x}, \mathbf{y}_c) + \log p(\mathbf{x}, \mathbf{y}_c). \tag{19}$$

Eq.(18) follows by definition of the KL divergence, while Eq.(19) substitutes the result from the identification adjustments for the causal query $p(\mathbf{z}|\mathbf{x}, \hat{\mathbf{y}}_c)$ in Eq.(15). Based on Eq(19), the ELBO can be written as in Eq.(20), completing its derivation. Note that in Eq.(20) we have ommited the presence of a factor $\mathbf{u}_x$ and focused only on the observed factors $\mathbf{y}_c$.

$$\log p(\mathbf{x}^{(i)}, \mathbf{y}_c^{(i)}) \geq \mathcal{L}(\boldsymbol{\theta}, \phi; \mathbf{x}^{(i)}, \mathbf{y}_c^{(i)})$$

$$= \mathop{\mathbb{E}}_{q(\mathbf{z}|\mathbf{x}^{(i)}, \mathbf{y}_c^{(i)})} \left\{ \log p(\mathbf{x}^{(i)}|\mathbf{z}) \right\} - D_{KL}\left( q(\mathbf{z}|\mathbf{x}^{(i)}, \mathbf{y}_c^{(i)})|| \mathop{\mathbb{E}}_{\mathbf{y}_{-c}} [p(\mathbf{z}|\mathbf{y})] \right). \tag{20}$$

## D Benchmark Experiments

### D.1 Generating correlations in benchmark dataset

Correlations in the generated data where produced by the method described in Träuble et al. (2021); Roth et al. (2022) with $\sigma$ quantifying the amount of correlation between factors. The smaller the $\sigma$, the stronger the correlation is, and vice versa. All pair-wise correlations where generated with $\sigma = 0.1$, while a $\sigma = 0.2$ was used to generate the factor correlated with all others (i.e., 1-to-all), in consistency with Träuble et al. (2021); Roth et al. (2022).

### D.2 DL model settings

The VAE architectures used throughout the benchmarking experiments follows the implementations of Locatello et al. (2020); Roth et al. (2022). The encoder consists of 2x [Conv(32,4,4) + ReLU], 2x [Conv(64,4,4) + ReLU], MLP(256), MLP(2x10). The Decoder uses: MLP(256), 2 x [upConv(64,4,4) + ReLU], 2 x [upConv(32,4,4) + ReLU], [upConv(3,4,4) + ReLU]. Inputs are images with 3 channels grouped into batches of 64 images. Training is performed using the Adam optimizer with a learning rate of 10e-4 for 300,000 training steps. In the case of Factor-VAE, the architecture includes six layers of [MLP(1000), leakyReLU] followed by an MLP(2).

In terms of the functional encoder/decoder approximators, deep model capacity is assumed to satisfy the data processing inequality with equality constraints. In other words, the mutual information $I$

between $\mathbf{Y}_c$ and $\mathbf{Z}$ is preserved relative to $\mathbf{Y}_c$ and $\mathbf{X}$ (i.e. $I(\mathbf{Y}_c, \mathbf{Z}) = I(\mathbf{Y}_c, \mathbf{X})$). This assumption has been used in other works Locatello et al. (2020); Mao et al. (2022) and justified in the VAE's objective to faithfully approximate the marginal data distribution.

### D.3 DCI

The DCI disentanglement metric Eastwood & Williams (2018) is a measure of how each variable (or dimension) captures at most one generative factor. It can be computed for each variable or dimension $i$ as $D_i = (1 - H_K(P_i))$. Here, $H_K(P_i)$ is entropy given as $H_K(P_i)) = -\sum_{k=0}^{K-1} P_{ik} \log_K P_{ik}$ and $P_{ij} = R_{ij}/\sum_{k=0}^{K-1} R_{ik}$ is the probability of a learned latent variable being important for predicting a known generating factor. This later (i.e.,$R_{ij}$ ) can be computed from the classification prediction error.

## E  EXPERIMENTS WITH CHEMCAM REAL-WORLD DATASET

### E.1  DATASET DETAILS

The ChemCam LIBS instrument Wiens et al. (2012) datasets contain raw and denoised spectra obtained from a variety of targets (e.g., rocks, soil) and from reference calibration standards of known and certified chemical composition. The specific datasets we employ consists of spectrally resolved LIBS signal measurements collected on Earth in a laboratory setting from a set of $\sim 585$ reference calibration standards Clegg et al. (2017) and on Mars from a set of 10 reference standards of known true composition. Each target is repeatedly shot (e.g., 50 times) following each time measurements of the full 240-905 nm LIBS signal. After collection, wavelengths within the bands [240.811,246.635], [338.457,340.797], [382.13,387.859], [473.184,492.427], [849,905.574] were ignored out consistent with practices of the ChemCam team Clegg et al. (2017).

### E.2  TRAINING AND IMPLEMENTATION DETAILS

Hyperparameters of the DL model were set to an initial lr of 1.0, decayed after 75 epochs with cosine annealing Loshchilov & Hutter (2017) and with #epochs 300. Batches were constructed at each epoch from a set of $64$ shot-averaged examples randomized over the whole training set without replacement. The shot-averages where computed by averaging the LIBS signal representations over an individual target and laser shot location. This averaging is consistent with common practices of the ChemCam team Wiens et al. (2013); Clegg et al. (2017). From a practical standpoint, regularization by ReI in Eq.(5) requires computing expectation over distributions of the generative factors. This is computationally intractable and we resorted to approximations by sampling with a limited number of samples (throughout the experiments with spectral data we used a 1000 samples) per causal relationship. This approximation resulted in some information leaks from other generating variables. This phenomenon can be observed qualitatively for example in the small peaks present in Fig.2c (from 200-500nm wavelengths).

### E.3  ADDITIONAL COMPARISONS AGAINST DL ARCHITECTURES AND DEPTHS

Here, we include the results of a few additional experiments in the ChemCam dataset that compare performance on out-of-distribution examples. Table 4 provides additional results comparing performance on Earth-to-Mars transfer on a variety of DL architectures and averaged over all elements $\mathbf{y} \in \mathbb{R}^n$ with $n = 11$.

Comparisons include fully connected (FC), multilayer perceptron (MLP), MLP Mixer Tolstikhin et al. (2021), ResNet He et al. (2016), U-Net Ronneberger et al. (2015), Transformers Dosovitskiy et al. (2020), VAE Kingma & Welling (2013), $\beta$-VAE Higgins et al. (2016), Factor-VAE Kim & Mnih (2018), DIP-VAE Kumar et al. (2018). Note that some of the architectures do not produce a latent representation explicitly, these are however rather trained end-to-end for prediction. The number in parenthesis next to each architecture name (e.g., FC(10)) expresses the corresponding depth of layers. The results of Table 4 show that VAE+ReI outperforms standard architectures in cases of OOD examples regardless of the inductive biases implied by the compared architectural designs. The unsupervised representation learning methods Beta-VAE, factorized VAE and DIP-VAE trained with

Table 4: Transfer Performance Comparison

| Architecture | RMSE (% oxide) |
|---|---|
| FC(10) | 5.19 |
| MLP(10) | 5.06 |
| MLPMixer(8) | 5.23 |
| ResNet(18) + FC(10) | 6.23 |
| U-Net +FC(5) | 5.12 |
| Transformer + FC(10) | 6.12 |
| VAE + FC(10) | 4.51 |
| $\beta$-VAE +FC(10) | 4.13 |
| F-VAE +FC(10) | 4.01 |
| DIP-VAE +FC(10) | 4.67 |
| ReI-VAE+FC(1) | **2.45** |

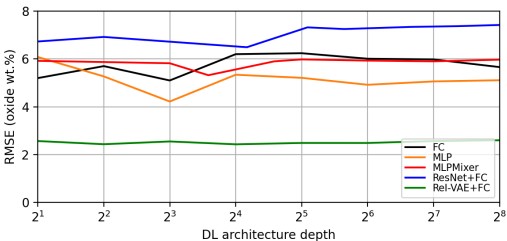

Figure 6: Transfer performance versus DL model depth.

a supervised prediction loss performed better at transfer than the standard deep learning architectures compared. However, VAE+ReI imposing disentanglement constraints via causal identification from the explicit DAG collider model, was able to outperform them all. Fig.6 also shows the transfer performance as a function of DL model depth. In this case, the FC, MLP, MLPMixer and ResNet+FC networks were compared. This plot shows that VAE+ReI is capable of outperforming standard DL models which did not exhibit generalization capabilities to OOD cases regardless of depth in this case. As a remark we would like to highlight that gains in task performance may not necessarily translate into more generalizable DL models. As evidenced by experiments, these may sometimes trick one's belief of a better model. In our case, these issues were settled through experiments evaluating the alignment of the resulting learned representations with domain knowledge. Finally, with regards to limitations, ReI requires a full reformulation of the learning problem when the data generation process is different from that of Fig.1. This human exercise of modeling the generation process through DAGs and deriving the conditions for identification of the causal effects can be time consuming. Discovering models of the generation process automatically Glymour et al. (2019) is an active area of research but this is outside the scope of this work. In some cases, causal identification for a given DAG can be more challenging to obtain or does not exist due to the presence of unobserved variables. Measurable proxies can be exploited as in Kuroki & Pearl (2014) in some of these cases, but in some others where this is not possible one has to resort to parametrization approximations which may result in entanglements of residuals between the true and sampled parameterized distributions; this, of course without identification guarantees. In the example application of chemical composition from LIBS we discussed this issue in the case of the sensor noise factor, with Wiens et al. (2013); Castorena et al. (2021) and without control as shown in Figs.2f and 2c, respectively..

We conclude this subsection by highlighting an additional significant drawback of the state of the art methods for disentanglement in comparison to ours: they do not produce representations that align with domain knowledge. This limitation carries significant drawbacks specially in high-risk applications. It also extends to other fields where the need for highly interpretable models is paramount, such as scientific research. In these contexts, the ability to understand and interpret the underlying factors driving model predictions is crucial for making informed decisions and ensuring the reliability and safety of the outcomes. Addressing this limitation becomes particularly vital in such applications.

