# OpenReview forum: "Representation Disentanglement via Regularization by Causal Identification"
_ICLR.cc/2024/Conference — Submitted to ICLR 2024_

### Official Review · Reviewer_osMn · 2023-10-30

**Soundness:** 3 good
**Presentation:** 4 excellent
**Contribution:** 3 good
**Rating:** 6
**Confidence:** 4

**Summary:**

The paper proposes to use causal identification as a regularizer for solving the problem that modern disentanglement models are usually generating entangled factors. The regularizer is actually a measurement for dependence between factors by a combination of d-separation constraints. Several experiments support the main claim of the paper.

**Strengths:**

1. The proposed regularizer by a viewpoint of d-separation is novel.
2. The experiments are thorough.

**Weaknesses:**

1. The theoretical completeness and computational efficiency of the proposed method need more justifications.
2. The universality of so-called collider-bias should be stated more clearly.

**Questions:**

1. The proposition 1 should be mathematically more rigorous so that it is a self-contained theorem. What is " appropriate set"? The same problem applies to prop 3.
2. The definition of semi- in Fig 2. It seems that the two graphs are just to DAGs. Is there a definition that distinguish this case of "semi" with ""two different generative DAGs"?
3. It would be better if existence of collider-bias in real data can be quantitatively analyzed. How important it is if this problem is solved? Also, the evaluation of the part of "transfer" is slightly weak compared to others.

---

> ### Author Response · Authors · 2023-11-23
>
> The appropriate set is a set of variables that blocks or separates paths through control by conditioning on variable in those paths. There are graphical rules which specify which variables should be controled  to block dependencies and which ones should be left without control. These are the rules of the do-calculus at the heart of causal inference. We included in definition 2 the general set of rules are and in the Appendix the more formal 3 rules of the do-calculus which detail the graphical criteria to define those an appropriate set depending on the graphical structure of the DAG.
>
> For the sake of clarity, we have removed the DAG corresponding to the semi-Markovian model. The semi-Markovian DAG, models double headed arrows when there is uncertainty about the direction of causality between variables. We decided to drop this from the paper. At first we thought such semi markovian model would help illustrate that dependencies between the generating variables could be either way. However, as this picture is not clearly conveying that, we decided to drop it from the paper. Instead, we went ahead and explained how the Markovian collider structure can explain correlations between generating variables when a common effect is observed (e.g., conditional associations). In other words, when the common effect is observed, knowledge of one variable "explains away" the other.
>
> The problems associated with quantifying collider-bias are similar to those of sampling selection bias. Namely, it undermines external validity and reduces generalization to shifts in the target population. In that context, reducing this problem is not only helpful, but fundamental towards generalization. There are results about prediction and transfer in the main body of the paper, illustrating a performance comparison between in distribution and out of distribution for a variety of elements in the application problem of chemical element prediction from spectral signals using a real dataset. In the Appendix, we had included the average over all chemical elements ( total=11), and have also compared this against other architectures at a variety of neural network depths, including the $\beta$-VAE, Factor-VAE, DIP-VAE which are among the best performing methods used for disentanglement.

---

### Official Review · Reviewer_JUu3 · 2023-10-31

**Soundness:** 2 fair
**Presentation:** 1 poor
**Contribution:** 2 fair
**Rating:** 3
**Confidence:** 4

**Summary:**

This paper explores a causal approach of learning disentangled representations through a concept called Regularization by Identification (ReI). The motivation stems from the fact that existing approaches mistakenly find correlations between generative factors due to collider bias. The approach involves training a VAE with an extra regularization term aimed at enforcing a conditional independence constraint between the generative factors. Experimental results show that the proposed approach achieves better disentanglement results than alternatives.

**Strengths:**

Exploiting collider structure in representation learning is an interesting and novel concept. The experimental results are also very extensive and quite convincing.

**Weaknesses:**

The paper is poorly written. Here are some of the confusing points:
1. What are “generative factors” as discussed in Prop. 1? Are they in the dataset? If so, what is the difference between $y$ and $x$?
2. What does “proposition” mean in this paper? Are they claims with proofs? They seem to simply be declaring or defining something.
3. In Prop. 2, what does it mean to say that “the identification of the causal effect $p(x \mid do(y_c))$ provides the recipe to control for dependencies between the generating factors producing entanglements through a combination of d-separation constraints and data from $p(x, y)$”? Further, does this imply that $p(x \mid do(y_c))$ is the desired query of interest? And is the discussion about d-separating $X$ from $Y$, or is it about d-separating some variables of $Y$ from other variables of $Y$?
4. How exactly is ReI defined (as in Prop. 3)? Is it the second term of Eq. 2? This could be clarified better.
5. What is the motivation of having the objective of $p(z \mid x, do(y)) = p(x \mid z)p(z \mid do(x))$? How was this derived? If Bayes rule is used here, doesn’t that drop $p(x \mid do(y))$? Also, these terms seem very different from what was introduced earlier.
6. Is Prop. 4 an assumption? This has to be clarified.
7. What exactly is the issue of collider bias with other approaches, and how does this approach avoid it?

Some factual issues are highlighted below. There may be more that I did not catch due to the clarity issues from above.
1. The factorization in Eq. 1 is not enough to argue that a DAG is causal. For example, the graph $X \rightarrow Y$ factorizes as $P(X, Y) = P(Y \mid X)P(X)$. However, the same equality holds even if the graph is $X \leftarrow Y$, despite having a different causal interpretation. It must be clarified that the arrows of the DAG represent causal relationships.
2. Def. 3 is not quite accurate. Some causal queries can be identified even without a  d-separating set.

For these reasons, I cannot recommend acceptance of this paper.

**Questions:**

See weaknesses.

---

> ### Author Response · Authors · 2023-11-23
>
> We assume there is an underlying data generative process, where the data is generated by factors explaining variability. This assumption is typical in disentanglement methods (e.g., [Higgins et al., 2016, Burgess et al., 2018, Kim and Mnih, 2018, Khemakem et al., 2020, Locatello et al., 2019] ) and methods vary depending on additional assumptions about such factors (e.g., i.i.d.), sparse support, etc. As used in this paper, variable $\mathbf{x}$ can represents signal measurements (e.g., 1D signals, images), variable $\mathbf{y}$ can represent labels.
>
> We have dropped the proposition declarations in the paper for the sake of clarity and because our main contributions are mainly models:  1) causal collider model to represent the underlying data generation model, regularization by causal identification model to impose constraints that controls dependencies between generating variables and $d$-separation given a graphical causal model and data as the criteria to test and control for entanglement between the generating variables.
>
> It means that we compute the causal effect of the generating factor $\mathbf{y}_c$ on signal/image $\mathbf{x}$. Causal effects are not biased, dependent or have influence from other generating variables outside from those in query (e.g., $\mathbf{y}_c$, $\mathbf{x}$).  In our case, we are using a causal graphical model which enables identifying causal effects (i.e., predicting the effect of interventions) from the DAG encoding the data generating process and data. The reason why we can do this, is because of the probabilistic implications of d-separation. Namely, that d-separation implies conditional independence, given the d-separating set $\mathbf{Z}$, in all distributions compatible with the assumptions encoded in the DAG.  If we can identify the causal effect from the DAG, then we can use the available data drawn from the joint distribution over the observations to compute the causal effect of an intervention on $\mathbf{y}_c$. Yes, the discussion deals with causal effect from $\mathbf{y}_c$. If the causal effect is identifiable then we can use it to learn representations disentangled relative to all other factors given the assumed DAG. Since identification of causal effects controls for any dependencies encoded in the DAG outside the query $p(\mathbf{x}| do(\mathbf{y}_c) )$, then this implies the separation of the generating factors. Moreover, as we are interested in learning disentangled representations of the generative factors from the observations $\mathbf{x}$ (i.e., signal, image ), it makes more sense to identify for $p( \mathbf{x} | do(\mathbf{y}_c)  )$, rather than $p( \mathbf{y}_j| {do(\mathbf{y}_c}) ) $ with $j \neq c$.
>
> Regularization by Identification is defined by the second term in Eq.2, while this term is derived by identification of the effect of the generating factors in the observed data point. In an attempt to clarify such point, we have modified that section where the definition was included.
>
> The posterior term $p(\mathbf{z} | \mathbf{x}, do(  \mathbf{y}_c)  )$ is motivated from variational inference. Derivation of the ELBO in the variational autoencoder is derived starting from that posterior. Denoising diffusion also uses a form of the ELBO to derive the posteriors at each time step. It is derived from Bayes rule, with some differences to manipulate interventions using the do-calculus. We have made some modifications to the paper and to its derivation to make this simpler to the reader, as included in the modified Appendix.
>
> Yes, we are assuming a collider-structured data generative model. We have dropped the propositions throughout the manuscript and rather just mentioned what we are proposing without a declaration distinction.

---

> > ### Author Response · Authors · 2023-11-23
> >
> > There are conditional associations between the underlying generating factors when the common effect is observed (e.g,, image, signal, video), even when the generating factors are independent. The collider structure graphical model under the causal lens explains such associations. Disentanglement methods in the literature have explained such correlations due to the presence of a confounding variables (e.g., [Lu et al;. 2021] and some have even disregarded the presence of such associations assuming independence between factors (e.g., Beta-VAE [Higgins et al., 2016], Factor-VAE [Kim & Mnih, 2018], Annealed VAE [Burgess et al., 2018],  etc). Although, models working under such assumption have shown to work very well in toy synthetic dataset problems where such an assumption can be controlled, such methods have difficulties in settings where independence does not hold [Trauble et al., 2021, Roth et al., 2022]. Both of these works [Trauble et al., 2021, Roth et al., 2022] go a step further and assume data can be generated by correlated factors, but without explaining the underlying cause. Here, we address such limitation, and propose a simple, but generalizable collider structure model that explains the presence of conditional correlations between the generating variables even when these are in reality independent. Our experiments provide evidence of collider bias and the performance results on both synthetic and real datasets support the idea that control of this collider bias through d-separation can help disentanglement.
> >
> > The factorization in eq. 1 is not a sufficient condition for the DAG to be causal and we did not mean this. Rather it is a property, one looks for when encoding the data generation process through a DAG. Causal inference relies on such DAG's along with graphical criteria tools (d-separation, rules of do-calculus) to predict causal effects, or rephrased in another way,  prediction of interventional probabilities; in contrast to standard probabilities of observations. The assumptions encoded in a DAG, remain as assumptions until the tools of causal analysis are applied to identify causal effects.
> >
> > Definition 3 has been modified in the paper. Causal identification is now written as a relaxed condition with an if statement only. We, in addition included the excellent reference [Ayem et al., 2023], which overviews causal effect identification methods.
> >
> > References:
> > [Ayem et al., 2023] Ayem, G. T., Thandekkattu, S. G., & Nsang, A. S. A Review of Causal Identifiability Techniques across Different Observational Datasets. International Journal of Current Science Research and Review. Vol 6, Issue 11, November 2023.
> > References
> >
> > [Burgess et al., 2018] Christopher P Burgess, Irina Higgins, Arka Pal, Loic Matthey, Nick Watters, Guillaume Desjardins, and Alexander Lerchner. Understanding disentangling in beta-vae. arXiv preprint arXiv:1804.03599, 2018.
> >
> > [Higgins et al., 2016] Irina Higgins, Loic Matthey, Arka Pal, Christopher Burgess, Xavier Glorot, Matthew Botvinick, Shakir Mohamed, and Alexander Lerchner. beta-vae: Learning basic visual concepts with a constrained variational framework. 2016.
> >
> > [Khemakem et al., 2020] Ilyes Khemakhem, Diederik Kingma, Ricardo Monti, and Aapo Hyvarinen. Variational autoencoders and nonlinear ica: A unifying framework. In International Conference on Artificial Intelligence and Statistics, pp. 2207–2217. PMLR, 2020.
> >
> > [Kim and Mnih, 2018] Hyunjik Kim and Andriy Mnih. Disentangling by factorising. In International Conference on Machine Learning, pp. 2649–2658. PMLR, 2018.
> >
> > [Locatello et al., 2019] Francesco Locatello, Stefan Bauer, Mario Lucic, Gunnar Raetsch, Sylvain Gelly, Bernhard Schölkopf, and Olivier Bachem. Challenging common assumptions in the unsupervised learning of disentangled representations. In international conference on machine learning, pp. 4114–4124. PMLR, 2019.
> >
> > [Lu et al., 2021] Lu, C., Wu, Y., Hernández-Lobato, J. M., & Schölkopf, B. (2021, October). Invariant causal representation learning for out-of-distribution generalization. In International Conference on Learning Representations.
> >
> > [Pearl 2009] Pearl, J. (2009). Causality. Cambridge university press.
> >
> > [Trauble et al., 2021] Frederik Träuble, Elliot Creager, Niki Kilbertus, Francesco Locatello, Andrea Dittadi, Anirudh Goyal, Bernhard Schölkopf, and Stefan Bauer. On disentangled representations learned from correlated data. In International Conference on Machine Learning, pp. 10401–10412. PMLR, 2021.

---

### Official Review · Reviewer_jyos · 2023-11-02

**Soundness:** 2 fair
**Presentation:** 1 poor
**Contribution:** 2 fair
**Rating:** 5
**Confidence:** 4

**Summary:**

the paper proposes a causal-inspired regularization term for representation learning. The innovation lies in the interventional distribution of the cause-effect and their identification condition, along with the proposed collider-based DAG model. An reformulation of VAE is presented with such DAG model, and empirical results show big improvements.

**Strengths:**

1. the idea of colider-based DAG model is new and authors show the motivation and connection with causal inference.
2. the empirical performance seems strong.

**Weaknesses:**

1. the major problem of the paper lies in the presentation. Many typo, inaccurate statements/notations, and grammatical errors exist. It hinders the understanding of the paper. For example,

- citation in latex is wrongly used.
- "are made up of causes of input and of outcome and " rephrase
- "independent of all its other predecessors" Markov definition is more general than discussed here. In addition, predecessors (and some other terms） are not defined formally.
"unbounded number of plausible models" the number of DAGs is bounded by the number of node in the graph?
- "by removing the effects of any non-causal dependencies between the input and outcome." what is an effect of non-causal dependencies?
- "While also"
-  graphical conditions: they don't seem to include cases where the path contains more than 3 nodes.
- "d-separation between the generative factors  $y_c \perp y_j | Z$ implies conditional independence" independence implies independence?
- The propositions are rather informally stated and contains incomplete statements. Prop 1: what is Z? Prop 2: what does "provides the recipe" mean?
- Prop 3: should it be definition instead. Prop 4: should be assumptions.
- p(z|x, do(y)) = p(x|z)p(z|do(y)) : it does not hold.
- Z_c: seems like a subset of Z, but also include U. Notation is quite confusing here.
- Eq 10: what is y_c here?

and many more.

2. Some baselines on causal representation learning is missing. For example,
"INVARIANT CAUSAL REPRESENTATION LEARNING FOR OUT-OF-DISTRIBUTION GENERALIZATION", Lu et al '22, and many baselines within it.

**Questions:**

- "encoder is set to produce latent vectors the same size as the inputs ": do you assume that the input and latent variables are within the same space?  can one choose a lower dimension?

---

> ### Author Response · Authors · 2023-11-22
>
> To Weakness point 1:  We have made significant efforts to improve the overall presentation of the paper. We diligently went over the entire paper and made modifications throughout to make it more concise. We additionally, corrected for typos and errors, re-arranged a few paragraphs, re-written many sentences for clarity, corrected grammatical errors, etc.
>
> We modified the sentence describing non-causal paths to:
> Non-causal paths, on the other hand, consist of a sequence of connections between variables that lack a direct cause-and-effect relationship and represent dependencies that may arise from non-causal influences. Examples include confounding from shared causes or collider-bias arising by conditioning on a common effect.
> Parents $pa_i$ and predecessors are nodes defined along the ordered arrows in the graph. For example, in $X \rightarrow Z \rightarrow Y $, $X$ is the only parent of $Z$, $Z$ is the parent of $Y$ and $\{X,Z\}$ is the list of predecessors of $Y$.
> We are referring to the unbounded number of plausible models in the context of a joint distribution. In other words, the number of models one can decompose a joint distribution (e.g., 1) $p(x,y,z) = p(y|z)p(z|x)p(x)$, 2) $p(y|x,z)p(z|x)p(x)$, 3) $p(x|z)p(z|y)p(y)$…and other ordered combinations ); this of course assumes there are no latent variables. The Markov compatibility property of DAG's restricts this number of plausible decompositions and the DAG encodes the measured and latent variables in the system. Note that this Markov compatibility assumes a first order Markov process. We have made such distinctions and included the additional definitions in the paper to improve clarity.
>
> Given a specific task (e.g., estimation, prediction, learning features),  the effect of non-causal dependencies (if untreated) between input and output is bias.
>
> "While also"… modified and corrected in the paper.
>
> Related to the graphical conditions. $X$, $Y$, $Z$ can represent sets of nodes, not exclusive to single nodes. We had specified "variable sets" in front of $X$ and $Y$ in the definition and same with $Z$. Such graphical conditions are for single nodes and generalizable to sets of nodes in consistency with Pearl [Pearl 2009].
>
> "d-separation between the generative factors $X \indep Y | \mathbf{Z}$ implies conditional independence" independence implies independence?. In the paper, we were previously using $X \indep Y | \mathbf{Z}$ to denote d-separation between variables $X$ and $Y$ given set $\mathbf{Z}$. This d-separation as used in the causal context corresponds to the graphical criteria which holds under the specified DAG. Such notation was consistent with Pearl's literature [Geiger et al., 1990, Pearl, 2009]. Note that this is different from $X  \indep Y | \mathbf{Z}$ describing the probabilistic notion of conditional independence. To clearly differentiate this notion from the probabilistic one (this notation ia also consistent with  [Pearl, 2009]), we now denote d-separation in the paper by $(X \indep Y | \mathbf{Z})_G$ and the probabilistic notion of conditional independence by $(X  \indep  Y | \mathbf{Z})_P$.
>
> $Z$ comes from the definition of d-separation. It is a set that blocks or d-separates pairs of single or sets of nodes. The recipe to control refers to the means to control for non-causal dependencies between input and output by conditioning on the set  $Z$ that d-separates the input and output. We merged both propositions 1 and 2 from the previous version and re-written it for clarity.
> In addition of providing the expected dependencies in the system through the DAG, d-separation allows establishing through graphical means the set of variables that need to be conditioned or control to control for those dependencies. By recipe we were referring to the how do we control for dependencies after finding the type of dependencies between variables in the system through the graphical criteria of d-separation.
>
> $p(z|x, do(y)) = p(x|z)p(z|do(y))$ : it does not hold. This was a typo, it should have been $p(z|x, do(y)) \propto p(x|z)p(z|do(y))$.
> $p(x|z, do(y)) = p(x|z)$ from Markov condition while $p(z,do(y)) = p(z|do(y)) \cdot p(do(y)) = p(z|do(y))$  follows as $p(do(y))=1$ (by definition of external intervention on a variable).

---

> ### Author Response · Authors · 2023-11-22
>
> With $Z_c$ we wanted to make it clear that it is the set that d-separates $y_c$ from the factors $y_c$. However, as you note this notation is not clear. We dropped the $Z_c$ notation and just refer to as the set $Z$, in general, Of course this set can take several instances depending on d-separation for specific variables. For example for $y_c$, $Z={y_{-c}}$, in other words, all factors excluding $y_c$.
>
> $y_c$ is $y_1$ here. For the sake of consistency we have modified notation in the appendix to make it consistent throughout the entire manuscript.
>
> Thank you for pointing out the paper from [Lu et al., 2021]. We have now referred this paper in our work. We already had a few of the referred works in our paper (e.g., [Suter et al., 2019, Khemakem et al., 2020]) and added a few more that we were unaware of. We would like to point out that although disentanglement is related to OOD generalization, the main focus of our is the former where this task follows the standard baselines of the $\beta$-VAE [Higgins et al., 2016] , Factor-VAE [Kim & Mnih, 2018], Annealed VAE[Burgess et al., 2018],  including the causal relevant ones such as [Locatello et al., Trauble et al., 2021] and their associated standard DCI metrics [Eastwood & Williams, 2018]. These methods have standardized the use of the shapes3d, dSprites,MPI3D ML benchmark datasets for the specific task. On the OOD focused line of work the baselines included in [Lu et al, 2021], associated metrics and benchmark datasets are specifically suited for OOD generalization and not disentanglement. The methods described therein rely on dataset shifts from environment changes. Although this task is related to disentanglement, our primary focus is disentanglement.
>
> References
>
> [Burgess et al., 2018] Christopher P Burgess, Irina Higgins, Arka Pal, Loic Matthey, Nick Watters, Guillaume Desjardins, and Alexander Lerchner. Understanding disentangling in beta-vae. arXiv preprint arXiv:1804.03599, 2018.
>
> [Eastwood and Williams, 2018] Cian Eastwood and Christopher KI Williams. A framework for the quantitative evaluation of disentangled representations. In International Conference on Learning Representations, 2018.
>
> [Geiger et al., 1990] Dan Geiger, Thomas Verma, and Judea Pearl. d-separation: From theorems to algorithms. In Machine Intelligence and Pattern Recognition, volume 10, pp. 139–148. Elsevier, 1990.
>
> [Higgins et al., 2016] Irina Higgins, Loic Matthey, Arka Pal, Christopher Burgess, Xavier Glorot, Matthew Botvinick, Shakir Mohamed, and Alexander Lerchner. beta-vae: Learning basic visual concepts with a constrained variational framework. 2016.
>
> [Khemakem et al., 2020] Ilyes Khemakhem, Diederik Kingma, Ricardo Monti, and Aapo Hyvarinen. Variational autoencoders and nonlinear ica: A unifying framework. In International Conference on Artificial Intelligence and Statistics, pp. 2207–2217. PMLR, 2020.
>
> [Kim and Mnih, 2018] Hyunjik Kim and Andriy Mnih. Disentangling by factorising. In International Conference on Machine Learning, pp. 2649–2658. PMLR, 2018.
>
> [Locatello et al., 2019] Francesco Locatello, Stefan Bauer, Mario Lucic, Gunnar Raetsch, Sylvain Gelly, Bernhard Schölkopf, and Olivier Bachem. Challenging common assumptions in the unsupervised learning of disentangled representations. In international conference on machine learning, pp. 4114–4124. PMLR, 2019.
>
> [Lu et al., 2021] Lu, C., Wu, Y., Hernández-Lobato, J. M., & Schölkopf, B. (2021, October). Invariant causal representation learning for out-of-distribution generalization. In International Conference on Learning Representations.
>
> [Pearl 2009] Pearl, J. (2009). Causality. Cambridge university press.
>
> [Suter et al., 2019] Raphael Suter, Djordje Miladinovic, Bernhard Schölkopf, and Stefan Bauer. Robustly disentangled causal mechanisms: Validating deep representations for interventional robustness. In International Conference on Machine Learning, pp. 6056–6065. PMLR, 2019.
>
> [Trauble et al., 2021] Frederik Träuble, Elliot Creager, Niki Kilbertus, Francesco Locatello, Andrea Dittadi, Anirudh Goyal, Bernhard Schölkopf, and Stefan Bauer. On disentangled representations learned from correlated data. In International Conference on Machine Learning, pp. 10401–10412. PMLR, 2021.

---

### Official Review · Reviewer_o5A3 · 2023-11-10

**Soundness:** 2 fair
**Presentation:** 3 good
**Contribution:** 2 fair
**Rating:** 5
**Confidence:** 3

**Summary:**

In this work, the authors propose regularization by identification (ReI), an approach to obtain disentangled representations by imposing generative factor disentanglement constraints through causal identification. The authors note the collider bias seen in other approaches to obtain disentangled representations and propose this regularization mechanism to overcome the collider bias. This is achieved by describing disentanglement in terms of d-separation in a directed acyclic graph (DAG). Moreover, the authors provide a reformulation of the VAE that adds ReI regularization to the ELBO which keeps the likelihood term intact. Finally, the effects of ReI in removing the effects of collider bias and obtaining disentangled representations are shown in disentanglement benchmarks and a real-world dataset.

**Strengths:**

- The paper presents a novel approach to obtain disentangled representations. The motivation behind removing the effects of collider bias is clearly explained. The mathematical fundamentals are clearly introduced and motivated with simple examples (though the presentation could further be improved by clarifying the notation used towards the beginning, as the variables sometimes switch during the explanations without a clear framework).
- Applying the method to the real-world dataset obtained from LIBS shows its robustness to OOD samples. Moreover, the fact that ReI could be introduced to the VAE regularizer given by the KL divergence without affecting the likelihood ensures the separability of the two, paving the way for ReI to be applied to additional frameworks, which could be of interest to the community.

**Weaknesses:**

- It is not clear how the DAG characterizing the causality can be obtained, as it is assumed to be given based on my understanding. In particular, for more complex examples, identifying causal and non-causal dependencies based on the generative factors seems particularly difficult. The same applies to supervisory signals. This is currently the main weakness in the approach in my view.
- In the DCI comparisons, the paper compares the proposed VAE+ReI method with related works such as the $\beta$-VAE. However, as mentioned in the paper, the $\beta$ scalar in $\beta$-VAE (and related approaches compared in the experiments) controls the strength of enforcing the latent prior such that the DCI scores might heavily depend on the selected value. It would be good to see the values selected for the different datasets and the effect of selecting a few different scores on the results (or at least to know that the selected $\beta$ was the best-performing one from a set of values). In addition, it would be good to see the standard deviation of the metrics for the 10 seeds.

**Questions:**

- As mentioned in the paper, d-separation between the generative factors implies their conditional independence given the set that d-separates them. The absence of d-separation, on the other hand, implies a dependence in almost all distributions compatible with the DAG. When is the dependence not implied through the absence of d-separation and what consequences would it have for the possibility to learn disentangled representations?

---

> ### Author Response · Authors · 2023-11-22
>
> To Reviewer o5A3
> Weakness point 1:
> We are providing the DAG of the underlying data generation process for the disentanglement problem. Here below we include a few steps that outline how to obtain a DAG for a general application problem. This is from Pearl's perspective and the following steps summarized by Boris Sobolev in one of his courses.
> 	1. Add input and output nodes (e.g., image "X" and labels "Y")
> 	2. Add nodes of factors that influence the output and draw directed arrows towards the output node.
> 	3. Add nodes of factors that influence the input and draw directed arrows towards the input node.
> 	4. Merge the nodes in steps 2 and 3 that are the same.
> 	5. Add nodes for factors that change in response to input and draw directed arrows from input to these factors.
> 	6. If any of the factors added in Step 5 also influence the output, draw directed arrows from these to the output node.
> 	7. If any of the factors added in Step 5 are influenced by the output, draw directed arrows from the output node to these nodes.
> 	8. Add nodes for intermediary factors between already connected nodes for known mechanisms
> 	9. Add directed arrows establishing directional dependencies between the nodes added in step 8.
>
> Things we would like to note here is the DAG explicitly summarizes the assumptions (e.g., dependencies between variables, conditional independencies, etc) made in the problem. Such assumptions are open for rigorous critic by the domain experts. The tools to establish causal and non-causal dependencies between variables are not so hard to use, either. In essence, they are described by the direction of arrows (which encodes influence) between input and output variables. There is graphical criteria summarized as the d-separation criterion and more comprehensively as the rules of the do-calculus which contains the axioms to test and control for causal and non-causal dependencies between variables. These are at the core in causal inference from [Pearl, 2009]. Other more complex methods to obtain the DAG consist on discovery [Glymour et al., 2019] or learning [Guo et al., 2020], these are however outside the scope of our paper.
>
> We would like to also point out that explicitly encoding our assumptions about a specific problem can help, specially in cases when datasets do not necessarily conform to the i.i.d. setting. DAG's can explicitly summarize the data generation process in datasets to indicate datasets with plausible selection bias, datasets with cycles, heterogeneous/nonstationary variables, datasets with confounding, datasets with missing values [Ayem et al., 2023] and in general point out differences between training population and target population. Modern data-based methods whose aim is to construct models that will predict an outcome from the input variables, relying on the assumption that the data properly represents the population of interest, are intrinsically fragile. In terms of supervisory signals, our work is limited by the availability of some form of supervision.
>
> To Weakness point 2:
> We refer the reader to the study by [Locatello et al., 2019] where the authors conduct a comprehensive study including the effect of regularization strength $\beta$ against performance. In our experiments, we used their disentanglement_lib implementation code along with the included guidelines as provided therein. In particular, the $\beta$ value was swept over the set [1, 2, 4, 6, 8, 16, 20, 24] and the findings show that the empirical best performing Beta is at 16  [18.3, 29.2,41.6, 51.4, 56.2, 70.3, 61.2, 54.9]. However, these, vary depending on the dataset.  For the experiments conducted in the ML benchmark datasets, we used the values of: (12) dSprites, (16) Shapes3D, and (8) MPI3D . In general as the $\beta$ strength is increased results in a decrease in correlation between generating factors, this comes however at a cost of data fidelity. The later produced as the KL term involved in the VAE is promoted over the likelihood term as the regularization strength increases. We would like to also note that such experiments are consistent with comparisons made in the works of [Roth et al. 2022] for the same benchmark datasets experimented here.
>
> We have also modified tables I,II,III in the paper to include the resulting standard deviation of the 10 seeds. The standard deviation shows consistency with the percentile values found in [Roth et al., 2022]. Comparisons of the mean and standard deviation values demonstrates the superiority of regularization through ReI with the collider structure relative to the compared disentanglement methods.

---

> ### Author Response · Authors · 2023-11-22
>
> Reply to Questions:
> Dependence is not implied by the absence of d-separation when there are influences between variables in the underlying data generation process which where not encoded in the DAG. In other words, when there is model mismatch between the causal graph model and the underlying data generation process due to for example missing of important variables and/or edge connections or by edge connections that do not respect the direction of influence between variables according to the underlying data generation process.
>
> References:
>
> [Ayem et al., 2023] Ayem, G. T., Thandekkattu, S. G., & Nsang, A. S. A Review of Causal Identifiability Techniques across Different Observational Datasets. International Journal of Current Science Research and Review. Vol 6, Issue 11, November 2023.
>
> [Glymour et al., 2019] Glymour, C., Zhang, K., & Spirtes, P. (2019). Review of causal discovery methods based on graphical models. Frontiers in genetics, 10, 524.
>
> [Guo et al., 2020] Guo, R., Cheng, L., Li, J., Hahn, P. R., & Liu, H. (2020). A survey of learning causality with data: Problems and methods. ACM Computing Surveys (CSUR), 53(4), 1-37.
>
> [Locatello et al., 2019] Locatello, F., Bauer, S., Lucic, M., Raetsch, G., Gelly, S., Schölkopf, B., & Bachem, O. (2019, May). Challenging common assumptions in the unsupervised learning of disentangled representations. In international conference on machine learning (pp. 4114-4124). PMLR.
>
> [Roth et al., 2022] Roth, K., Ibrahim, M., Akata, Z., Vincent, P., & Bouchacourt, D. (2022, September). Disentanglement of Correlated Factors via Hausdorff Factorized Support. In The Eleventh International Conference on Learning Representations.
> [Pearl 2009] Pearl, J. (2009). Causality. Cambridge university press.

---

### Meta-Review · Area_Chair_Ugbn · 2023-12-13

**Metareview:**

In this paper, the authors argue that deep representation learning models for disentanglement are ill-posed with collider bias behavior, which explains the conditional associations between the generating factors given their effect. They further propose regularization by identification (ReI), which is a modular regularization engine to align the behavior of large-scale models with the disentanglement constraints imposed by causal identification. Empirical results illustrate the effectiveness of ReI in removing the effects of collider bias.

While the paper presents a novel approach to achieving disentangled representations, this study and the presentation of the paper are to be improved in multiple ways. First, the paper compares the proposed VAE_ReI with baselines like beta-VAE; then the value of beta in beta-VAE will be important for a fair comparison. Several statements (including some definitions) should be technically clearer and more rigorous.

**Justification For Why Not Higher Score:**

This study and the presentation of the paper are to be improved in multiple ways. First, the paper compares the proposed VAE_ReI with baselines like beta-VAE; then the value of beta in beta-VAE will be important for a fair comparison. Several statements (including some definitions) should be technically clearer and more rigorous.

**Justification For Why Not Lower Score:**

The paper presents a novel approach to achieving disentangled representations.

---

### Decision · Program_Chairs · 2024-01-16

Reject